# Nanoparticles for Drug and Gene Delivery in Pediatric Brain Tumors’ Cancer Stem Cells: Current Knowledge and Future Perspectives

**DOI:** 10.3390/pharmaceutics15020505

**Published:** 2023-02-02

**Authors:** Luana Abballe, Zaira Spinello, Celeste Antonacci, Lucia Coppola, Ermanno Miele, Giuseppina Catanzaro, Evelina Miele

**Affiliations:** 1Department of Pediatric Hematology/Oncology and Cellular and Gene Therapy, Bambino Gesù Children’s Hospital, IRCCS, 00165 Rome, Italy; 2Department of Experimental Medicine, Sapienza University of Rome, 00161 Rome, Italy; 3Cavendish Laboratory, Department of Physics, University of Cambridge, Cambridge CB3 0H3, UK

**Keywords:** nanoparticles, nanodelivery systems, pediatric brain tumors, cancer stem cells

## Abstract

Primary malignant brain tumors are the most common solid neoplasm in childhood. Despite recent advances, many children affected by aggressive or metastatic brain tumors still present poor prognosis, therefore the development of more effective therapies is urgent. Cancer stem cells (CSCs) have been discovered and isolated in both pediatric and adult patients with brain tumors (e.g., medulloblastoma, gliomas and ependymoma). CSCs are a small clonal population of cancer cells responsible for brain tumor initiation, maintenance and progression, displaying resistance to conventional anticancer therapies. CSCs are characterized by a specific repertoire of surface markers and intracellular specific pathways. These unique features of CSCs biology offer the opportunity to build therapeutic approaches to specifically target these cells in the complex tumor bulk. Treatment of pediatric brain tumors with classical chemotherapeutic regimen poses challenges both for tumor location and for the presence of the blood–brain barrier (BBB). Lastly, the application of chemotherapy to a developing brain is followed by long-term *sequelae*, especially on cognitive abilities. Novel avenues are emerging in the therapeutic panorama taking advantage of nanomedicine. In this review we will summarize nanoparticle-based approaches and the efficacy that NPs have intrinsically demonstrated and how they are also decorated by biomolecules. Furthermore, we propose novel cargoes together with recent advances in nanoparticle design/synthesis with the final aim to specifically target the insidious CSCs population in the tumor bulk.

## 1. Introduction

Malignant brain tumors represent the leading cause of cancer-related deaths during childhood [1]. Advances in surgery, radiotherapy and chemotherapy have ameliorated the survival rate for some cancers, such as medulloblastoma (MB) and pediatric low-grade glioma (pLGG), but for diffuse intrinsic pontine glioma (DIPG) and high-grade gliomas (pHGG) the prognosis is still poor [2]. However, current therapies applied to pediatric tumors worsen patient’s quality of life and are associated with long-term sequelae in terms of endocrine, neurological and cognitive disorders [3,4]. Advances in molecular and gene profiling of brain tumors have improved diagnosis, risk stratification and identification of aberrant genetic pathways, allowing us to appreciate differences with adult tumors and paving the way for new “personalized” treatment modalities. Nonetheless, among the novel therapeutic approaches targeting gene mutations and dysregulated pathways, as well as the harnessing of immunosurveillance and immune system, vaccine therapy and virotherapies are still limited. For immunotherapies, such as CAR T-cell and virotherapies, intrinsic limitations are few targetable tumor antigens, insufficient proliferation or expansion and lack of durable response [5,6]. The use of nucleic acids with therapeutic purpose is an interesting platform because of their low cost and simple synthesis. For instance, silencing and halting the expression of multiple oncogenes can be achieved with therapeutic tools based on siRNAs/CRISPR/Cas9. On the other hand, miRNA delivery in tumor cells gives the advantage to perturb with a single-molecule multiple crucial pathway. However, applications of nucleic acid-based therapies are still in the early stages due to concerns such as the lack of stability, off-target effects observed and poor cellular uptake. Furthermore, depending on the route of administration, the presence of nucleases in the serum reduces the bioavailability and accumulation of an efficient dose of “drug” in the brain [7]. The lack of clinical studies devoted to pediatric patients and the need for low toxicity and high therapeutic effect demonstrate that there is an urgent need to look for specific, effective delivery methods [8]. The accessibility of brain tumors poses other layers of complexity due to the anatomic location and the presence of the blood–brain barrier (BBB) which impedes and controls drug penetration and accumulation. The impermeability of the BBB is due to specialized endothelial cells, pericytes and astrocytes [9]. Tight junctions and efflux pumps contribute to the impermeability and the selectivity of substrates across the barrier, blocking the accumulation of drugs in the brain parenchyma [9]. In addition, in cancer diseases, the BBB undergoes modifications in terms of the integrity, vascularization and efflux pump expression [10]. Cellular heterogeneity represents an additional challenge for the realization of a successful therapeutic strategy [11]. Indeed, the heterogeneous stage of differentiation of cells in the tumor bulk makes the design of a targeted therapy more challenging. Cancer stem cells (CSCs) have been identified in different types of pediatric brain tumors, such as medulloblastoma, ependymoma and gliomas [12,13]. CSCs are a subpopulation of cells defined by their self-renewal capacity, differentiation properties and functionally by the ability to form a tumor mass when engrafted in an immune deficient mouse [14]. Importantly, CSCs have been extensively described as resistant to radiotherapy and chemotherapy and thus actively contribute to tumor recurrence and poor outcomes after treatment [15,16]. High-throughput single-cell analyses have revealed that CSCs in highly heterogeneous brain tumors, such as medulloblastoma and high-grade glioma, exhibit broad plasticity. CSCs’ plasticity is the most widely accepted theory for explaining tumor heterogeneity in glioblastoma (GBM). According to this model, CSCs can differentiate into tumor cells while also returning to an undifferentiated state in response to cell-intrinsic (genetic and epigenetic factors) and cell-extrinsic factors (TME influences), resulting in a dynamic and heterogeneous tumor mass [17] (Figure 1).

Epithelial–mesenchymal transition (EMT) is one of the hallmarks of cellular plasticity; different subsets of CSCs, depending on their spatial location in the TME, exhibit different EMT phenotypes [18]. Moreover, TME signals (such as hypoxia) can also regulate EMT, favoring the most aggressive cell phenotype [19], via the generation of stem-like cells [19].

In the GBM context, CSCs perform a metabolic adaptation depending on available tumor microenvironment factors (oxygen and nutrient), supplied by surrounding non-neoplastic cells, to support tumor growth and stemness features [20] (Figure 1). CSCs responding to environmental cues also change their phenotypic state, such as modulating the expression of cell membrane markers, which has important implications for stem cell surface marker-based target therapy design [17,21,22].

In the last years, an application of nanoparticles (NPs) in cancer therapy has emerged [23]. The term NPs refers to physical objects with characteristic length in the range 1–100 nm [24]. Nanomaterials mostly investigated in brain tumor research are characterized by different physico-chemical composition: organic based (e.g., polymers, dendrimers, liposomes), inorganic or made of more than one material [25]. NPs can be loaded with drugs and employed as a drug delivery system (DDS) [26] for different cargoes, such as proteins, peptides and nucleic acids. The attention on NPs’ application for brain tumor treatment is due to their potential to overcome biological barriers, such as the BBB [27]. Indeed, several efforts have been performed to modify and enrich NPs with moieties capable to better interact with both the BBB and the blood tumor–brain barrier to specifically reach and discriminate cancer cells. Another interesting aspect is the application of NPs as a diagnostic tool. Indeed, NPs can be used as theranostic agents, in a synergistic manner, to deliver therapeutic and imaging agents together [28,29]. Therapeutic strategies based on NPs would benefit from their intrinsic ability to access the niche where the CSCs reside and successfully act on the crosstalk between the tumor and microenvironment, as well as target the unique phenotypes and functional properties of CSCs. Several studies have proved the great potential of NPs’ functionalization to specifically target these cells in tumors of different tissue origins such as in breast, colon, prostate, melanoma, leukemia, pancreas and adult brain tumors. The efficacy of conventional chemotherapeutics has been improved via the delivery of NPs in both in vitro and in vivo models [30]. The possibility to “customize” the NPs’ properties and cargoes enables us to target the unique biological features of CSCs, to avoid the tumor recurrence and to increase the survival rate of children affected by brain tumors. In this review, we will summarize the current state-of-art in nanoparticle-based therapy of pediatric brain tumors. We will focus on the stem cell compartment and propose relevant cargoes to specifically target these cells.

## 2. Nanoparticles and Target Therapy

### 2.1. Synthesis of NPs

NPs used in cancer therapy can be obtained from organic and inorganic materials or by a combination of them. Besides the “raw” material employed, NPs can differ in size, structure and shape. These features can be tuned and depend on the approach used in the synthesis method employed. NPs can be obtained by using the top-down or the bottom-up approach. The top-down method starts with bulk material to obtain smaller units by disruption or decomposition that are then converted into NPs. Conversely, the bottom-up approach involves atom materials that are progressively “clustered” and then converted into NPs (Figure 2).

In both approaches, the synthesis is achieved using physical methods (such as mechanical milling, thermal decomposition, spinning), laser method (ablation, pyrolysis) or by using chemical techniques [31]. Research is moving towards novel methods and sources of synthesis, such as biosynthesis via bacteria, fungi and plants. The green synthesis is an eco-friendly and low-cost method that reduces the risk and the toxicity connected with NPs synthesis and application [32,33]. The combination of organic and inorganic materials along with post-synthesis functionalization make them suitable for DDS. Indeed, chemical properties of NPs ensure the stability of drugs and biomolecules, the evasion from uptake, enzymatic and immune clearance. Surface modification of NPs helps to overcome biological barriers and local drug-loading. Functionalization of NPs with antibodies or aptamers specifically addresses drugs to tumor cells avoiding normal bystander cells.

### 2.2. Mechanisms of Action

A crucial aspect of NPs is their ability to specifically target cancer cells, which improves therapeutic effectiveness while avoiding damage to normal cells. Two are the main mechanisms used by NPs to reach target sites: passive targeting and active targeting.

### 2.3. Passive Targeting

For the first time, in 1986, Hiroshi Maeda and colleagues described the pathophysiological phenomenon that occurs in solid tumor vasculature known as the “enhanced permeability and retention effect” (EPR effect) [34]. This mechanism describes the intrinsic ability of macromolecules to reach and accumulate in the solid tumors’ interstitium, based on tumor pathophysiological characteristics such as: (i) neovascularization, characterized by deficient basement membranes and fenestrated structures of endothelial tubes, (ii) upregulation of inflammatory factors and (iii) lack of efficient drainage of lymphatic systems, that together sustain the delivery, accumulation and retention of molecules into solid tumor tissues [35,36]. Passive targeting exploits the EPR effect on the delivery and retention of drugs at target site. However, in clinical settings, this strategy has not always worked as well as hoped due to a variety of factors, including the tumor type, location, blood perfusion status, physical-chemical characteristics of delivered agents and difficulty in predicting the distribution of drugs.

### 2.4. Active Targeting

The concept of active targeting is based on the direct interaction between ligands and receptors. This interaction is exploited by the active targeting of NPs on cancer cells, with high affinity and precision, reducing, on one hand, the cytotoxic effects on non-target cells, and, on the other hand, favoring the endocytosis by tumor cells [37,38]. NPs can be functionalized through the conjugation of ligands on their surfaces. They can be “decorated” with (i) monoclonal antibodies or their fragments against receptors or surface molecules over-expressed in target cells, (ii) proteins or peptide-based molecules, (iii) nucleic acids, and (iv) small molecules. After the ligand-receptor binding, NPs are internalized via receptor-mediated endocytosis and they can successfully release the drugs inside target cells [39]. To overcome the limitations of drug delivery in brain tumors, linked to the presence of BBB, many receptors have been examined for their role in crossing BBB. To improve NPs’ ability to transport molecules across the barrier, NPs have been functionalized against receptors expressed on BBB cells. Here, we report the best-studied and promising receptors for brain tumor delivery, including the epidermal growth factor receptor (EGFR), transferrin receptor (TfR), insulin receptor (IGFR) and lipoproteins (Figure 3).

#### 2.4.1. Epidermal Growth Factor Receptor

EGFR is a tyrosine kinase receptor (RTK) that interacts with a variety of EGFR family ligands regulating various aspects of cell growth and development. In the context of human brain tumors, EGFR is overexpressed in a subset of GBM and MB, and EGFR targeting is a promising approach for the NPs’ delivery [40,41].

EGFR-decorated NPs (EGFR-NPs) can be used as carriers for chemotherapy agents that are not able to cross the BBB, in physiological conditions (such as temozolomide, TMZ), and that are subsequently internalized by target cells via receptor-mediated endocytosis. Several studies reported examples of EGFR-NPs strategies in glioma tumors. Liu and colleagues developed drug-loaded NPs functionalized with dual-targeting of EGFR (expressed on tumor cells) and of low-density lipoprotein receptor-relative protein-1 (LRP1) (expressed on the BBB endothelial). The dual targeting exhibited enhanced BBB penetrability and tumor targeting effects, in both in vitro and in vivo glioma models [42]. Another study [43] reported the efficiency of EGFR-NPs for the delivery of photosensitizer silicon phthalocyanine (Pc 4) against glioma tumors. Additionally, Whittle et al. described a phase I clinical trial on patients affected by recurrent GBM treated with weekly administration of novel nano cellular compounds loaded with a chemotherapeutic agent and functionalized with anti-EGFR antibodies, demonstrating no dose-limiting toxicity in fourteen patients [44].

Beyond the drug delivery, EGFR is an attractive biomarker also for brain tumor imaging, acting as both a diagnostic and therapeutic agent. This method could be used for the real-time tracking of NPs in crossing the BBB, in the accumulation of NPs within target tissues and in providing shape contour-defining imaging of the tumor. This strategy also aids in the detection of distant metastasis and residual disease following incomplete surgical resection. Additionally, the measurement of the dilution of NPs could be used as a marker of the cell proliferation rate in the different areas of the tumor. Among imaging agents, fluorescent molecules are gaining popularity in clinical diagnostics [45,46]. Hadjipanayis et al. reported a study in which magnetic NPs conjugated to an EGFR deletion mutant (EGFRvIII) antibody are used for GBM detection (via magnetic resonance imaging (MRI)) and targeted therapy [47] (see also Table 1).

#### 2.4.2. Transferrin Receptor

The Tf receptor (TfR) is a transmembrane glycoprotein with two subunits linked by a disulfide bridge, each of which can bind to a molecule of transferrin, and functions to transport iron into cells. TfRs are widely expressed in the body, including red blood cells, hepatocytes, monocytes, erythrocytes, intestinal cells and normal brain cells (endothelial, neurons and glial cells) but also in brain cancers, making TfR an excellent candidate for the design of targeted therapy in brain tumors [48,49]. One of the advantages of using Tf-conjugated nanoparticles (Tf-NPs) is the ability to cross the BBB. However, one of the limitations is that the supplied Tf, carried by Tf-NPs, must compete with the endogenous Tf in plasma, and the BBB TfRs are >99.9% saturated with endogenous Tf [50,51]. Another limitation is that the exogenous Tf could lead to an overdose of the iron transport into the brain, NPs conjugated to TfR-targeted antibodies are preferable to Tf-NPs because they bind a different site on TfR [52]. GBM cells overexpress TfRs to respond to the increased demand of iron to sustain tumor growth, the expression of TfRs in this tumor is indeed almost 100-fold higher compared to healthy normal astrocytes. Many studies reported the use of Tf in drug delivery approaches for GBM therapy [53,54,55,56,57,58]. TfR is also over-expressed on the surfaces of GBM cancer stem cells (GSCs), so it could be considered a common target for both stem and bulk tumor cells. Based on these observations, Sun and colleagues developed a nano-strategy, using TMZ-loaded Tf-NPs, able to effectively penetrate the BBB and target both tumor compartments: glioma stem cells and non-stem cells [59]. Also, Kim et colleagues have developed anti-cancer NPs, using TfR as a common target, in both CSC and non-CSC populations [60] (see also Table 1).

#### 2.4.3. Insulin Receptor 

Insulin was the first molecular “Trojan horse” described able to cross BBB and to deliver somatostatin (Pardridge, W.M. Chimeric Peptides for Neuropeptide Delivery through the blood–brain barrier. US Patent 4,801,575, 31 January 1989). Subsequently, Shilo and colleagues developed insulin-targeted NPs and demonstrated, in in vivo models (male BALB/c mice), the ability to cross the BBB five times more than controls [61]. Furthermore, 83-14 monoclonal antibody to the human insulin receptor was used to functionalize NPs which had a greater ability to cross the BBB with respect to the anti-TfR antibody [62]. The anti-insulin receptor antibody 83-14 was also successfully used by Dieu et al., who demonstrated in in vitro experiments the specific endocytosis of these NPs by brain endothelial cells [63]. In another study, insulin or an anti-insulin receptor monoclonal antibody (29B4) linked to NPs were found to be capable of transporting non-penetrating drugs across the BBB [64] (see also Table 1).

#### 2.4.4. Lipoprotein

The low-density lipoprotein (LDL) receptor (LDLR) is over-expressed in tumors (cells and blood vessels) and in the BBB, which allows the use of LDL-NPs for brain-targeted therapy. Various in vivo and in vitro studies reported an increased diffusion of LDLR ligands-functionalized NPs: LDL or Apolipoprotein E and B (ApoE and ApoB) [65,66]. A study led by Grafals-Ruiz et al. highlighted the potential for the use of ApoE-conjugated-NPs in GBM tumors, in in vitro (U87 GBM cells) and in vivo models (GBM syngeneic mice) [67]. Also, synthetic nano-LDL particles were proposed as effective drug delivery vehicles for GBM [68,69]. Additionally, NPs conjugated to Angiopep-2, which is a ligand that binds to the LDL low-density lipoprotein receptor-related protein (LRP), was proposed as an excellent option for drug delivery, as described by Kadari et al. in human (U87MG) and mouse (GL261) glioma cell lines and mouse GBM models [70]. Of note, high-density lipoprotein nanoparticles (HDL NPs) have intrinsic anti-tumoral activity in targeting the cholesterol signaling pathway. HDL-mimetic NPs were successfully used by Bell et al. in Sonic Hedgehog (SHH)-driven MB, demonstrating the high-affinity of HDL NPs in binding the HDL scavenger receptor type B1, SCARB1, depriving cells of natural HDL and cholesterol cargo. This strategy resulted in a promising approach, highly dependent on cellular cholesterol levels, to target stem cell compartments [71] (see also Table 1).

## 3. Nanoparticles and Brain CSCs Compartment

MB is the most common malignant pediatric brain tumor arising in the cerebellum [72,73]. Current treatments are mainly directed to target the bulk cancer cell population, and thus have little effect on CSCs. Emerging effort was given to the development of the specific nano-delivery of therapeutic/tracing molecules to the CSC niche, taking advantage of the presence of surface markers or sites of overexpressed enzymes (World Health Organization. WHO Report on Cancer: Setting Priorities, Investing Wisely and Providing Care for All. WHO; Geneva, Switzerland: 2020). However, cell nanotechnology also faces many challenges for targeting CSCs such as: the identification of specific surface markers, the anatomic location of cancer stem cells (NPs are not able to reach the hypoxic central core, where blood flow is impaired) and the microenvironment (differences in the environmental pH and temperature between tumor and healthy tissues) [74]. Nowadays, NPs have been used to improve several aspects of tumor management, such as safety, efficacy and in vivo tracking. 

The main approach used to target the CSCs is the use of NPs-encapsulated drugs functionalized with antibodies/ligands targeting simultaneously two or more CSCs markers. Kim et al. developed high-density lipoprotein-mimetic nanoparticles (eHNPs) incorporated with apolipoprotein A1 (to enhance the BBB crossing) and anti-CD15 (a murine SHH MB cancer stem-like cell target) to achieve dual targeting and a SHH inhibitor (Sonidegib (LDE225)) to affect CSCs. eHNPs effectively crossed the BBB, delivered drugs and had powerful anti-neoplastic activity on the CSC population, demonstrated in different SHH MB models: cells in vitro, ex vivo in organotypic tumor slice culture and in vivo (model of a SmoA1 MB tumor-bearing mouse) [23]. Also, Bell et al. employed eHNPs targeting the scavenger receptor class B type 1 (SCARB 1), via binding to the HDL receptor, in SHH-MB. This strategy deprived tumor cells of their natural HDL and cholesterol cargo and depleted the CSCs population [71]. Lim and colleagues used NP-loaded curcumin to treat the SHH-MB cell line, DAOY [75]. Several studies suggested that curcumin exerted anti-tumoral effects on CSCs impairing self-renewal and clonogenicity [76] and that NPs-curcumin were able to target MB CSCs, decreasing clonogenic growth, the stem cell receptor CD133 positive CSCs and attenuated two of the main stem cell survival signaling pathways: the STAT3 and Sonic Hedgehog. 

Among gliomas, pediatric-type diffuse high-grade gliomas (pHGG) are malignant tumors which arise from different brain locations, such as the cerebral hemispheres, thalamus, brain stem or spinal cord [15]. The existence of CSCs was reported firstly for adult glioblastoma (GBM) [77] and then described in pHGG [13]. Current therapies specifically targeting GSCs affect signaling pathways and metabolism. NP applications and studies are mainly focused on adult gliomas, such as GBM. Few studies applying the NP approach were conducted specifically on pHGG. A study conducted in DIPG, an aggressive pediatric brain tumor, tested in vitro on tumor neurospheres the serum albumin coated passion-fruit-like nanoarchitectures (NAs-HSA) loaded with the chemotherapeutic agent doxorubicin. Doxorubicin and doxorubicin-loaded NAs-HSA had a similar effect on DIPG neurospheres. Colony formation assays demonstrated greater potency of NAs-HSA-Dox on inhibiting colony formation compared to doxorubicin [78]. Shargh et al. developed three nanodelivery systems of TMZ analogue (N3Propargyl) including an apoferritin (AFt) nanocage, a sulfobutyl ether β-cyclodextrin (SBE-β-CD) nanocomplex, and SBE-β-CD in nanoliposomes and tested on 2D-spheroids and 3D obtained from DIPG cell lines resistant to TMZ. The formulations were efficacious in reducing the tumor sphere size [79]. 

## 4. Relevant Cargoes for Brain CSCs

NPs have many applications in medicine, but the main role in cancer treatment is certainly complex molecule delivery. We listed the principal cargoes, with different molecular and physico-chemical properties, in the brain CSCs cancer stem cells target strategy (see also Table 2).

### 4.1. Drugs

Nano-delivery protects drugs from rapid degradation, reducing the effective therapeutic dose by minimizing the side effects of its intrinsic cytotoxicity. On the other hand, it carries the drug into the TME tumor microenvironment and promotes cellular uptake. The most common NPs-transported drugs in brain tumors are presented below. According to Infante et al., in the context of SHH-MB CSCs, NPs can be used to overcome the drugs’ poor water solubility. Glabrescione B (Gla B), an SHH inhibitor, was delivered in a self-assembling amphiphilic polymer micelle. This strategy demonstrated high drug loading and stability, low cytotoxicity, and tumor growth inhibition in allograft and orthotopic models of Hh-dependent MB [80]. Kim J et al. [81] developed engineered eHNPs composed of apolipoprotein A1 and CD15 (a murine SHH MB cancer stem-like cell target) to selectively deliver the Smo inhibitor, sonidegib (LDE-225) in the SHH MB tumor site. In vitro, in vivo and ex vivo experiments demonstrated that these engineered biomimetic NPs are effectively able to cross the BBB and deliver drug molecules to cancer stem-like cell populations, enhancing the therapeutic efficacy of LDE-225 in SHH MB treatment. Bukchin et al. described a nano-strategy to encapsulate and deliver the topoisomerase I (TOP1) inhibitor SN-38. The authors used patient-derived DIPG cell models, both as monolayers and cultured as tumor spheres in a tumor stem medium. SN-38 displayed a very low aqueous solubility profile and does not cross the BBB that in this tumor is well-conserved, but it is strongly efficient against DIPG cells in vitro. To increase CNS bioavailability, the authors designed SN-38 loaded NPs of amphiphilic chitosan (CS)-g-poly (methyl methacrylate)-poly (acrylic acid) copolymer that was surface-modified with peptide to improve BBB transport. Their results confirmed the anti-DIPG efficacy of SN-38 NPs to kill almost 85% of stem-enriched DIPG cells and to successfully cross the BBB endothelium (in vitro) [82]. A dual targeting of CSCs niche (via curcumin) and bulk tumor cells (via doxorubicin) were developed by Xu and colleagues in the context of a glioma tumor. Briefly, doxorubicin (DOX) hydrochloride and curcumin (Cur), which are characterized by differences in pharmacokinetics and BBB permeability, were co-delivered via pH-sensitive core-shell NPs. 

Two different methods of NP encapsulation were used based on the drugs’ properties: hydrophobic Cur was loaded in the cationic core, whereas hydrophilic DOX was in the outer shell polymer of NPs through electrostatic interactions. DOX and CUR were effectively delivered in vivo in a rat model of glioma and decreased the percentage of CSCs from 4.16% to 0.9%. This type of NP may be a promising strategy for delivering drugs with different physicochemical properties [83]. In another study, Kim JS et al. [84] have designed a strategy to improve TMZ delivery to the stem cell compartment in a GBM orthotopic mouse model by conjugating a TMZ-encapsulating immuno-liposome with Angiopep-2 (ANG) mAB (for BBB transcytosis) and CD133 mAB (for specifically target GBM stem cells).

As mentioned above, receptors overexpressed via cancer cells can be loaded on NPs for targeted therapy. CSCs possess receptors on their surface that have been used as markers to isolate and identify this rare population. The unique immunophenotype of CSCs permits their discrimination among the other cells. A typical marker of brain tumor stem cells is CD133, a glycoprotein promoting tumorigenesis, with spheroid formation ability. Several strategies to specifically target CD133 have been explored, including the use of antibodies, aptamers and immunotherapy [85]. Aptamers have emerged as suitable moieties to be loaded onto NPs [86]. These molecules are advantageous for their small size, granting permeability across the barrier compared to other molecules, such as antibodies. Several aptamers have been proposed to specifically target CSCs and hence are seen as potential candidates to load on NPs. Recently, a 15-nucleotide base-pair aptamer targeting CD133 was loaded on Au-PEG NPs to deliver the drug Telaglenastat (CB-839). Au-PEG-CD133-CB-839 particles exerted a strong decrease in the GBM cell lines’ survival compared to controls [87]. Affinito et al. described an RNA aptamer that selectively targets the Ephrin A-2 on the cell surface of GCSs, inhibiting tumor stemness and migration [86]. 

NPs loaded with drugs represent the main types of NPs currently tested in clinical trials for pediatric brain tumors. Indeed, the clinical translation of nanomedicines in this field is full of challenges and obstacles due to [88]: (i) the heterogeneity of NPs’ composition and cargoes, making it hard to define the most effective NP in terms of formulation, doses and route of administration; (ii) the lack of predictive in vitro and in vivo models; (iii) limitations in technical and analytical methods to measure brain uptake and nanotoxicity; (iv) the absence of clear nanomedicine regulations. For these reasons, the encapsulation of already-approved drugs into NPs provides a way to overcome, at least in part, these issues. Four clinical trials involving the use of NPs loaded with drugs are ongoing in pediatric brain tumors: doxorubicin-loaded liposomes [NCT00019630, Phase 1, pediatric brain tumors], cytarabine-loaded liposomes [NCT00003073, Phase 1, pediatric brain tumors] and panobinostat NP formulation [NCT03566199, Phase 1, DIPG; NCT04264143, Phase 1, DIPG]. In addition to these, three other clinical trials have been activated: two of them concern the use of NPs as a diagnostic tool [NCT00978562, Early Phase 1, pediatric brain tumors; NCT00659334, Phase 2, pediatric brain tumors], while the other tests NPs for palliative treatment [NCT03250520, Early Phase 1, pediatric brain stem gliomas].

### 4.2. Small Interfering RNA

Small interfering RNA (siRNA)-based strategy silencing technology is an interesting tool to target proteins that are crucial in cancer diseases. Ideal target gene candidates in cancer therapy comprise oncogenes, proliferation, angiogenesis and cell cycle [89].

Delivery of siRNA to the brain is challenging due to the low stability of the molecule and the accessibility of brain parenchyma [90]. The naked siRNA molecules delivered through NPs successfully reach the brain tissue. 

The use of NPs loaded with siRNA is reported in pediatric ependymoma and MB in an in vitro study. PEGylated chitosan-modified gold NPs were employed to deliver siRNA against Ape1 enzyme, which repairs DNA damage and sustains cancer cells upon damage induced via gamma irradiation [91]. Cohen et al. demonstrated the delivery of RNAi polo-like kinase 1 (PLK1) to chemo-resistant grade IV glioma. SiRNA was directed to the GBM site using hyaluronan (HA)-grafted lipid-based nanoparticles (LNPs). The LNP’s surface was functionalized with HA, which is glycosaminoglycan that recognizes the stem cell receptor CD44, expressed on GBM cells. HA-LNPs loaded with polo-like kinase 1 (PLK1) siRNAs (siPLK1) downregulated the expression of PLK1 mRNA and induced cell death. Furthermore, in a human GBM U87MG orthotopic xenograft model, the convection of Cy3-siRNA entrapped in HA-LNPs was performed and uptake was observed in U87MG cells, showing significantly prolonged survival of treated mice in the orthotopic model [92]. Yu. et al. have described a proof-of-concept for a multiplexed strategy of siRNA delivery applied on brain tumor-initiating cells (BTICs) and in vivo in an established mouse tumor. SiRNA knockdown of crucial proneural transcription factors, namely SOX2, OLIG2, SALL2 and POU3F2, were delivered using 7C1- lipopolymeric nanoparticles NPs (LPNPs) [93]. This formulation strongly reduced the expression of siRNA-targeted proteins and reduced the spherogenic potential. Furthermore, the BTIC xenograft and intratumoral delivery of siRNA encapsulating 7C1 NPs significantly reduced tumor growth.

**Table 2 pharmaceutics-15-00505-t002:** Examples of different types of NPs used in pediatric brain tumor models. List of NPs and cargoes used to target brain tumor CSCs.

Nanoparticle	Cargo	Application	Type of Cancer	Mechanism of Action	In Vitro/In Vivo Models	References
Self-assembling amphiphilic polymer forming micelles, called mPEG_5kDa_-cholane	Glabrescione B (Hedgehog inhibitor)	Drug delivery	SHH MB	Inhibition of tumor growth	- DAOY human MB cells; - Murine primary MB cells isolated from Math1-cre/PtcC/C mice; - Allograft and orthotopic models of Hh-dependent MB	[67]
HDL-mimetic nanoparticles (eHNPs) composed of apolipoprotein A1 and CD15	LDE-225 (Smo inhibitor, sonidegib)	Drug delivery	SHH MB	Intracellular cholesterol depletion and cytotoxicity	- DAOY human MB cells; - PZp53 cells; - SmoA1 Transgenic (Jax 008831); - Math-Cre-ER-Ptch flox/flox	[68]
Amphiphilic polymeric nanoparticles modified with a protease resistant peptide	SN-38 (Topoisomerase I inhibitor)	Drug delivery	DIPG	Apoptosis	- Patient-derived DIPG cell models; - Hsd: ICR mice	[69]
pH-sensitive core-shell nanoparticles	Doxorubicin hydrochloride and curcumin	Drug delivery	Glioma	Cytotoxicity	Rat model of glioma	[70]
Immunoliposome using angiopep-2 and anti-CD133 monoclonal antibody	Temozolomide	Drug delivery	GBM	Cytotoxicity and reduction of CD133+-positive cells	Glioblastoma orthotopic mouse model	[71]
CD133-Functionalized Gold Nanoparticles	GLS1 inhibitor Telaglenastat (CB-839)	Drug delivery	GBM	Inhibition of glutaminolysis	- GBM1 cells; - NCH644 cells; - LN229 cells; - U87 cells	[74]
Hyaluronan (HA)-grafted lipid-based nanoparticles	RNAi polo-like kinase 1 (PLK1)	Small interfering (siRNA) delivery	GBM	Inhibition of glutaminolysis	- GBM cell lines and primary neurosphers of GBM patients; - GBM U87MG orthotopic xenograft model	[78]
Lipopolymeric nanoparticle	Multiple siRNAs (SOX2, OLIG2, SALL2 and POU3F2)	Small interfering (siRNA) delivery	GBM	Inhibition of self-renewal and tumorigenicity	- Patient-derived GBM cells (GBM6, GBM12, GBM26, GBM43, MGG8, MES83); - Patient-Derived Xenograft Mouse Model of GBM	[79]
Bioreducible poly(beta-amino ester) nanoparticles	miR-148a and miR-296-5p	microRNAs (miRNAs) delivery	GBM	Inhibition of tumorigenicity	- Human GBM derived neurospheres (GBM1A); - Orthotopic human GBM xenografts	[92]
Polyfunctional gold–iron oxide nanoparticles (polyGION)	miR-100, anti-miR-21 and temozolomide	microRNAs (miRNAs) and drug delivery	GBM	Increased in vivo survival	- U87-MG cells; - U87-MG GBM cell-derived orthotopic xenograft models	[93]
Serum albumin coated passion fruit-like nanoarchitectures (NAs-HSA)	Doxorubicin	Drug delivery	HGG	Apoptosis in vitro but not in vivo	- Primary DIPG cultures (HSJD-DIPG007, SU-DIPGVI, RA055 and VUMC-DIPG10) - Orthotopic DIPG Animal Model (HSJD-DIPG007 cells)	[65]

PEG: polyethylene glycol; SHH MB: Sonic Hedgehog medulloblastoma; Hh: Hedgehog; HDL: high density lipoprotein; DIPG: diffuse intrisic pontine glioma; GBM: glioblastoma; HGG: high grade glioma.

### 4.3. microRNAs

Non-coding RNAs are emerging as critical regulators of biological processes in brain cancers [94].

Among non-coding RNAs, microRNAs (miRNAs) have been identified as differentially expressed in pediatric brain tumors compared to normal/healthy brains, influencing survival and response to therapy. MiRNA signatures in pediatric brain tumors reveal how miRNA are differentially expressed in tumor with respect to normal tissue [95], in responders versus non responders to chemotherapy [96] and even the tumor anatomical location [97], indicating that some miRNA might be silenced. 

MiRNAs associated with CSCs, differentially expressed in six pediatric solid tumor cell lines have been identified as potential targets [98]. For instance, miR-584-5p was demonstrated to potentiate vincristine and radiation therapy in both in vitro and in vivo models of MB [99]. In MB downregulation of miR-125b, miR-324-5p and miR-326 could be associated with the regulation of SHH target genes, such as Smo, Gli1 and Ptch1 [100]. The miR-326 is downregulated in MB Stem-Cells (MBSCs) derived from SHH-MB, where it acts as a negative regulator of self-renewal [101]. Hes1, downstream Notch signaling, is crucial for MB CSCs maintenance and was found regulated via miR-199-5p [102]. Another study conducted on pHGG revealed that Notch2 activity is at least in part controlled by miR-107, miR-181 and miR-29a-3p, both by themselves and in a concerted manner [103]. Liang et al., in 2017, compared miRNomes and transcriptomes of pLGGs and pHGGs identifying miR-137 and miR-6500-3p downregulated in pHGGs [104]. Recently, miR 139-5p was found downregulated in supratentorial pLGGs. Notably, the overexpression of miR halted proliferation by controlling PI3K/AKT/mTOR signalling [105]. Additionally, miR-1248 has been described as a prognostic biomarker for disease progression in pLGGs [106].

Restoration or inhibition of the above-mentioned miRNAs to the physiological levels could be a therapeutic strategy used to quell survival strategies adopted specifically by CSCs inside the tumor bulk. However, miRNAs administration requires a delivery system that ensures the molecules’ stability. Lopez- Bertoni et al. took a multiplexed approach by designing a polymeric nanoparticle loaded with two miRNAs, miR148a and miR-296-5p, that could regulate tumor growth in GBM [106]. Poly-β amino ester particles used in the study were formulated to promote better gene delivery. Tumor-suppressing miRNAs delivered via NPs inhibited the growth of GBM xenografts and prolonged survival in mouse models [106]. A recent elegant study conducted by Sukumar and colleagues [107] proposed polyfunctional gold–iron oxide nanoparticles (polyGION) loaded both with miR-100 and an anti-miR-21 against GBM. miR-100 acts as an oncosuppressor, while miRNA-21 is dysregulated in GBM and contributes to gliomagenesis and therapy resistance. PolyGION NPs were first functionalized with chitosan-cyclodextrin (CD-CS) hybrid polymers to load negatively charged miRNAs via electrostatic interaction. NPs successfully deliver miRNA and anti-miRNA in GBM cells in vitro and sensitize cells to TMZ, inducing a decline in cell viability. Furthermore, the intranasal delivery of polyGION formulation loaded with miRNA and administration of TMZ increased the overall survival of mice.

## 5. Immunotherapy and NPs in Pediatric Brain Tumors

The complex interplay between cancer and immune cells regulates tumor development: indeed, immune system activity has a key role in controlling disease initiation and progression and at the same time the immune microenvironment is highly influenced by tumor signaling. For these reasons, immunotherapy represents a powerful therapeutic approach for many types of cancer but has shown less benefit against pediatric brain tumors. The presence of the BBB as well as the high tumor heterogeneity and a suppressive immune microenvironment have limited the development of effective immunotherapeutic approaches. To this purpose, combining immunotherapy with nanotechnology could provide novel opportunities to improve pediatric brain cancer therapy. Different types of nano-immunoconjugates have been developed to reduce immunosuppression or improve immune activation in brain tumor sites [108,109]. Checkpoint inhibitors, such as anti-CTLA-4 and anti-PD-1 [110] or anti-PDL-1 [111] have been covalently attached to NPs to activate an immune response in GBM; also cytokines have been loaded into NPs to treat GBM [112]. These approaches, despite being proven to be effective in modulating the immune system to treat brain cancers, can cause an over-activation of immune cells and off-target inflammation, making it necessary to develop alternative ways to affect immune cells. To this purpose, the therapeutic delivery of NPs encapsulating RNA seems to be a promising solution. Different types of RNA have been encapsulated in NPs and tested in several pediatric brain tumors, as follows: (i) mRNA encoding for transcriptions factors that induce inflammatory gene expression in immune cells in GBM [113]; (ii) siRNA to reduce immunosuppression by downregulating anti-inflammatory signals in GBM [114,115] and DIPG [116]; (iii) tumor-derived RNA that quickly activate a tumor-specific immune response in MB [117] and DIPG [118]. This last strategy seems to be the most translatable at a clinical level, as shown by the Food and Drug Administration Investigational New Drug (FDA-IND) approval for the first-in-human trials (IND#BB-19304) in pediatric patients with HGGs using RNA-NPs [PNOC020 study, NCT04573140]. Despite the fact that further investigations are needed to allow the use of nano-immunoconjugates in a clinical setting, this trial has paved the way for the development of NP-based personalized immunotherapy for pediatric brain cancer treatment.

## 6. Advanced Pre-Clinical Models to Study NPs in Pediatric Brain Tumors

The BBB represents one of the main challenges in brain cancer therapy due to its low permeability to the majority of drugs. As a result, pre-clinical studies are mainly based on the identification of strategies to enhance BBB-crossing and brain tumor accumulation of drugs. 2D cell cultures have been widely used to study drug responses in vitro, but they are not able to fully recapitulate the tumor features and microenvironment and to properly evaluate the therapeutic potential of anti-cancer compounds. For this reason, recent pre-clinical studies have been focused on the development of 3D culture models to better mimic the tumor’s complexity as well as BBB–brain tumor interactions and to properly predict the in vivo treatment responses. Perini G et al. [119] used a 3D brain cancer model to study the efficacy of liposomal TMZ precoated with a protein corona made of human plasma proteins in inhibiting tumor growth. The 3D culture was derived from the U87 human GBM cell line and spheroid size analysis and cell viability assays were performed to evaluate the anticancer activity of these NPs. The results showed that the treatment of 3D GBM culture causes a notable reduction of tumor size, in line with a considerable decrease in cell viability, demonstrating a marked inhibition of tumor growth. To mimic the BBB–GBM interaction in vitro, Straehla JP et al. [120] designed a microfluidic device of vascularized GBM using GBM spheroids, derived from the co-culture of the patient-derived xenograft (PDX) GBM22 cell line with pericytes (PCs), in direct contact with a self-assembled and perfusable vascular network made of induced pluripotent stem cells (IPS), human endothelial cells, astrocytes and pericytes. In this model, GBM spheroids grew rapidly in close contact with their surrounding BBB vasculature, similar to what can be observed in vivo in HGG patients and in animal models of GBM. The authors generated a NP composed of a liposomal core, coated with a first layer of poly-(l-arginine) and a second layer of propargyl-modified poly-(l-aspartic acid) (pPLD), and functionalized with angiopep-2(AP2) to target the GBM vessels via an interaction with the overexpressed BBB receptor LRP1. After demonstrating that targeted NPs are able to cross the BBB vessels near GBM tumor, the DNA-damaging agent CDDP was encapsulated in NPs and the therapeutical potential of these NPs was tested. The authors showed that NPs were able to accumulate in GBM spheroids and to increase the apoptosis of cancer cells with low damage to the surrounding healthy blood vessels. These results were confirmed in vivo in an orthoptic, intracranial murine tumor model where treatment with these NPs decreased tumor growth, demonstrating the reliability of this in vitro model as a pre-clinical model. Due to the crucial role of CSC population in the initiation and progression of cancer, Kim JS et al. [84] designed a strategy to improve NPs delivery to specifically target stem cell compartment in SHH MB in vitro, in vivo and ex vivo. After proving the ability of these NPs to cross the BBB and to accumulate in MB cells as well as the therapeutic effect on MB stem cells both in vitro (DAOY and PZp53 cell line) and in vivo (SmoA1-GFP-MB-bearing mouse model and patched (PTC) knockout model), they generated an ex vivo SmoA1 organotypic slice culture to better mimic the BBB–MB interactions and to evaluate the anti-cancer effect of these NPs in the tumor microenvironment. In slice cultures treated with eHNP-A1-CD15, the observed co-localization in the perivascular space of CD15 and NPs demonstrated that this treatment was able to specifically target the stem cell population. Furthermore, the staining for the cell death marker cleaved-caspase 3 (CC3) showed higher CC3 staining in slice culture treated with LDE225-loaded NPs, demonstrating the therapeutic efficacy of these compounds on MB-SHH.

## 7. Future Perspectives

Although drugs are the main tested and used cargoes in brain tumor nanotherapy, innovative approaches must be developed to increase drug delivery as well as to reduce toxicity; in this context, protein and ferroptosis therapies can be promising strategies. Protein therapy is a therapeutic approach that, due to the limited cellular uptake of the protein toxins, has the advantage to avoid the off-target cytotoxicity of chemotherapy agents. Jiang et al. [121] have generated an angiopep-2 (ANG)-directed and redox-responsive virus-mimicking polymersomes (ANG-PS) to chaperone saporin (SAP), a highly potent protein toxin. Interestingly, SAP, which is normally fast degraded in vivo and has poor cell permeability, when embedded in ANG-PS NPs, displays a high BBB transcytosis and GBM accumulation in vivo. Furthermore, the U87MG-Luc mouse model systemically treated with ANG-PS-SAP NPs shows a reduction of tumor growth and a longer median survival time with few adverse effects. Ferroptosis therapy is a new type of cancer therapy in which cancer cell death is induced by the production of ROS through an iron-based Fenton reaction [122,123]. This approach has high anticancer selectivity as the efficacy of this reaction relies on the local concentration of H+ and H_2_O_2_, particularly enriched in the TME. Shen Z et al. [123] have synthesized a novel kind of Fenton-reaction-acceleratable magnetic NP via the conjugation of lactoferrin (LF) and RGD dimer (RGD2), to enhance BBB transportability and cancer selectivity, with cisplatin (CDDP)-loaded Fe_3_O_4_/Gd_2_O_3_ hybrid NPs. The authors have demonstrated, in vitro and in vivo, that these NPs are able to cross the BBB and are internalized into cancer cells. In tumor cells they release Fe2+, Fe3+ and CDDP, that cooperate to give rise to the Fenton reaction, producing ROS and inducing cancer cell death. In the U87 MG orthotopic mouse model, a higher uptake of these NPs has been observed in the tumor site compared to the liver as well as an inhibition of tumor growth and extended survival time, demonstrating the efficacy of the ferroptosis therapy. In the last years, with the aim of improving the delivery of chemotherapy agents in brain cancers, NPs are widely used to build nanocomposite platforms and hydrogel that can be directly brain implanted. Scott et al. [124] have developed a microdevice made of poly(L-lactic acid) (PLLA) and a liquid crystal polymer (LCP) loaded with TMZ that, when intracranial implanted in 9 L gliosarcoma model in rodents, can locally release the drug over several days, resulting in a prolonged survival rate of treated animals. In a pilot study, Baltes S et al. [125] have loaded biocompatible sulfonate-modified polyvinyl alcohol (PVA) and hydrogel-based drug eluting beads (DEB) with DOX and irinotecan. After demonstrating the absence of local toxicity in healthy rats, the authors have shown that the intracerebral implantation of DEB loaded with DOX and irinotecan improves the survival time in a rat model of malignant glioma (8000 BT4Ca). Despite the promising results obtained in pre-clinical models, further investigations need to be conducted to evaluate if these strategies can be applied in a clinical context, allowing the development of new therapeutic approaches. CSCs are not isolated inside tumor bulk, since this has been extensively reported since they are surrounded by a complex TME [126,127]. 

Indeed, the brain CSCs’ fate is also determined by an intricate multicellular network (immune cells, stromal cells, blood and lymphatic vessels), extracellular matrix components and secreted factors. The crosstalk with TME components contributes to the maintenance of CSCs and the stimulation of tumor progression. Furthermore, the increase in energy demand of tumor bulk led to aberrant angiogenesis. Based on the technical advancements, the TME has been intensively analyzed because of transcriptomic, proteomic, metabolomic and spatial information. These innovative studies are giving new insights into how TME contributes to prognosis and therapy response [128,129]. Thus, therapies that use NPs are ongoing and are designed not only to disrupt the interplay of cancer cells with the surrounding TME, but also the remodeling of TME [130].

## 8. Conclusions

The complete therapeutic success against pediatric brain tumors is strongly hindered by the potential damage that therapies may induce on a developing brain. Prospective preclinical and clinical research must consider this aspect and therapy design should be intended to ensure efficacy and reduce side effects. The use of NPs as a delivery method of drugs and other bioactive molecules offers clear advantages in accessing the anatomic location of these aggressive cancers. As extensively described and reported, NPs can indeed overcome the BBB obstacle when decorated with receptors that are overexpressed both by BBB-building cells and cancer cells. This strategy allows discriminating cancer cells from surrounding healthy ones, thus directing the therapeutic agents towards malignant cells. However, tumor bulk is heterogeneity in cellular composition and the presence of CSCs, a less differentiated cellular population, cause augmented resistance to surgery and radio/chemotherapy. Thus, the specific eradication of the CSC compartment requires understanding and operating on the increasing amount of biological information on CSCs and their unique properties should be considered for the specific eradication of the CSC compartment. Different brain cancer pre-clinical models are used to evaluate the efficacy of designed NPs, however, preclinical studies have been conducted mostly on adult cancers, such as GBM. Further investigations on pre-clinical models designed specifically for pediatric brain tumors need to be developed. The improvement of NPs’ features and the delivery of chemotherapy drugs and other molecules have a huge potential for future investigations in the eradication of the CSC compartment. The crosstalk between TME and CSCs should be deeply studied. Achieving these objectives may have the potential to develop nanomedicine-based personalized therapy to treat pediatric brain cancer. (Figure 4).

The following key actors should be considered in a nano strategy to target brain CSCs such as: (i) NPs; (ii) the BBB; (iii) cancer stem cells; and (iv) pre-clinical models.

It is fundamental to consider all these complex actors to make possible the development of nanomedicine-based personalized therapy to treat pediatric brain cancer stem cells.

## Figures and Tables

**Figure 1 pharmaceutics-15-00505-f001:**
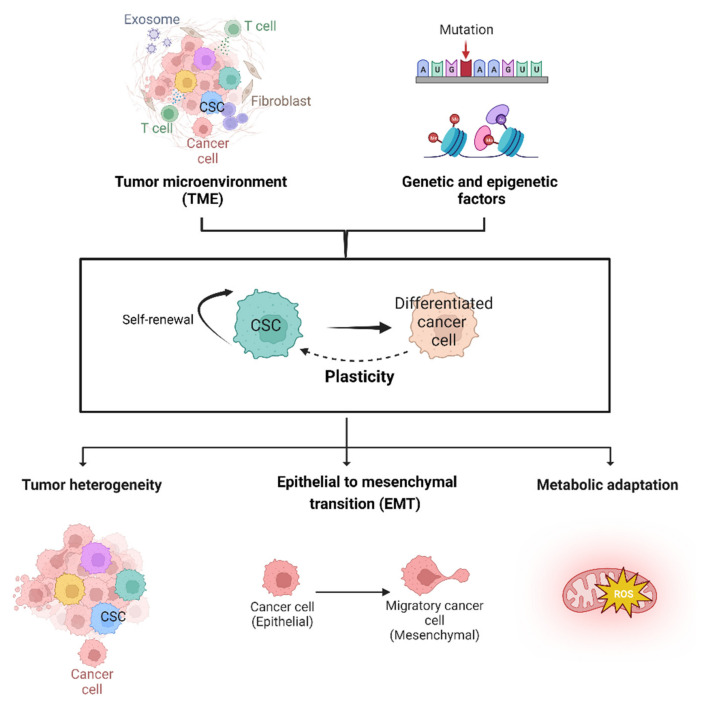
**Mechanisms involved in brain CSCs’ plasticity model.** CSCs plasticity is the capacity to shift dynamically and reversibly from dedifferentiated to a differentiated state. Brain CSCs plasticity depends on both tumor microenvironment factors (i.e., metabolites, cytokines, growth factors, exosomes) and genetic/epigenetics modifications. The main consequence of cell plasticity is the formation of highly heterogeneous tumor mass, characterized by the activation of EMT and metabolic adaption programs. Created with BioRender.com.

**Figure 2 pharmaceutics-15-00505-f002:**
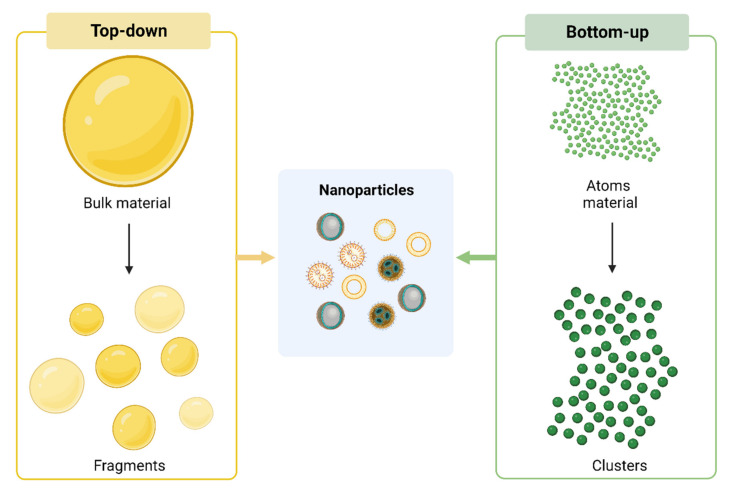
**NPs synthesis**. Current approaches employed for NPs synthesis: top down (on the left) and bottom-up (on the right). Created with BioRender.com.

**Figure 3 pharmaceutics-15-00505-f003:**
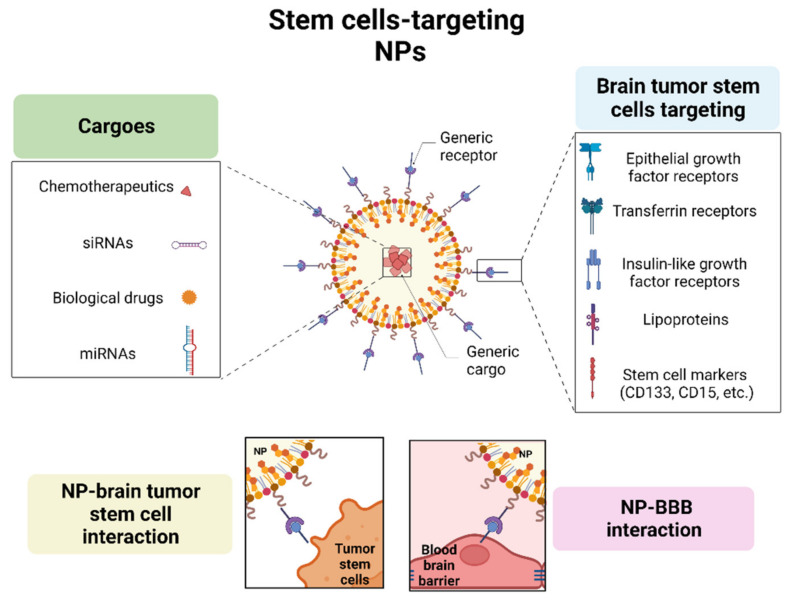
**NPs target therapy against brain cancer stem-cells.** Targeting: NPs platforms can be decorated with receptors that are overexpressed both by BBB and cancer cells (EGFR, TfR, IGFR, lipoproteins). Stem cell surface exposed markers (such as CD133, CD14) can be added to specifically reach brain CSCs. Cargoes: besides chemotherapeutics, NPs can deliver nucleic acids such as siRNA and miRNA in order to affect several aspects of cancer biology (such as modulation oncogenes/oncosuppressors). Using these strategies, NPs allow us to simultaneously go beyond BBB and specifically attack brain CSCs. Created with BioRender.com.

**Figure 4 pharmaceutics-15-00505-f004:**
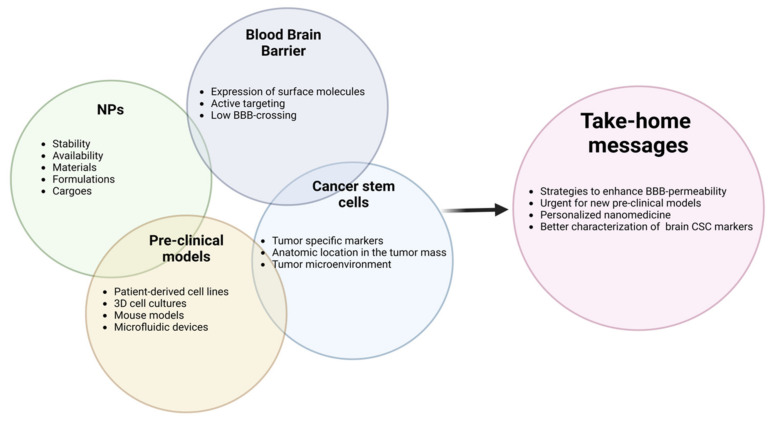
**Overview of the main aspects affecting nano-delivery strategy in pediatric brain cancer and future aims**.

**Table 1 pharmaceutics-15-00505-t001:** Example of receptors on tumor cells and BBB cells used in brain tumor nano approaches. List of promising receptors for brain tumors nano delivery.

Targeting Receptors	Type of NPs	Application	Target Cells	References
EGFR	NPs functionalized with Ang2 and EP-1	Drug delivery	Endothelial cells of BBB (Ang2) and tumor cells (EP-1)	[42]
	EGF-modified Au NP–Pc 4	Delivery of photosensitizer silicon phthalocyanine	Tumor cells	[30]
	NPs conjugated to an EGFR antibody (Panitumumab/Vectibix)	Drug delivery	Tumor cells	[31]
	Magnetic NPs conjugated to an EGFR deletion mutant (EGFRvIII) antibody	Magnetic resonance imaging	Tumor cells	[34]
TfR	Tf-conjugated nanoparticles	Drug delivery	Tumor cells	[53,54,55,56,57,58]
	Tf-conjugated nanoparticles	Drug delivery	Glioma stem cells and non-stem cells	[47]
	Tf-conjugated nanoparticles	Drug delivery	Glioma stem cells and non-stem cells	[53,54,55,56,57,58]
IGFR	NPs functionalized with anti-insulin receptor antibody 83-14	Drug delivery	BBB	[51]
	NPs functionalized with anti-insulin receptor monoclonal antibody (29B4)	Drug delivery	BBB	[52]
Lipoproteins	Gold-liposome nanoparticles conjugated with ApoE and RVG	RNAi delivery	Tumor cells (ApoE and RVG) and brain endothelium (RVG)	[67]
	Nano-LDL particles	Drug delivery	Tumor cells	[67]
	NPs conjugated to Angiopep-2	Drug delivery	BBB and tumor cells	[58]
	High-density lipoprotein nanoparticles	Intrinsic activity	MB cells and stem cells	[67]

NPs: nanoparticles; EGFR: epidermal growth factor receptor; Ang2: angiopep-2; BBB: blood–brain barrier; EP-1: EGFR-targeting peptide; EGF: epidermal growth factor; Pc: phthalocyanine; TfR: Tf receptor; IGFR: insulin growth factor receptor; ApoE: apolipoprotein E; LDL: low-density lipoprotein; MB: medulloblastoma; RVG: rabies virus glycoprotein; RNAi: RNA interference.

## Data Availability

Not applicable.

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
