# Peer review of "Nanoparticles for Drug and Gene Delivery in Pediatric Brain Tumors’ Cancer Stem Cells: Current Knowledge and Future Perspectives"

_pharmaceutics, 2023, doi:10.3390/pharmaceutics15020505_

Round 1
Reviewer 1 Report
Authors have reviewed nanoparticles-based drug and gene therapy for pediatric brain tumors. The review article is well-organized, and references are up to date. After careful reading the manuscript, I reached to a decision that the manuscript is suitable to be published by this journal subject to slight modification. My concerns are as follow:
1. Figure 1 is of low quality, please improve the figure quality for better readability.
2. On line 135, use the word 'successful'.
3. I would recommend adding figures corresponding to mechanisms of NPs targeting to brain tumors.
4. I would also recommend that authors add proper section of NPs synthesis and current advancements in synthesizing such nanocarriers with self-explanatory figures.
5. In the end, I also advise the authors to add a section of products that are under clinical trials or in pipeline. This would give a better idea of the success of NPs for targeting brain tumors in pediatric population.
Author Response
Reviewer #1
Authors have reviewed nanoparticles-based drug and gene therapy for pediatric brain tumors. The review article is well-organized, and references are up to date. After careful reading the manuscript, I reached to a decision that the manuscript is suitable to be published by this journal subject to slight modification. My concerns are as follow:
- Figure 1 is of low quality, please improve the figure quality for better readability.
Author’s reply: We modified the Figure 1 (now the New Figure 3) according to reviewer’ s suggestion.
- On line 135, use the word 'successful'.
Author’s reply: We corrected in the revised text (line 239).
- I would recommend adding figures corresponding to mechanisms of NPs targeting to brain tumors.
Author’s reply: We described the mechanism of how NPs target brain tumors in the New Figure 3, but we have modified the New Figure 3 to make it clearer.
- I would also recommend that authors add proper section of NPs synthesis and current advancements in synthesizing such nanocarriers with self-explanatory figures.
Author’s reply: We modified the text according to the reviewer suggestion, adding a new section “Synthesis of NPs” and a new Figure (Figure 2) (lines 149-177).
- In the end, I also advise the authors to add a section of products that are under clinical trials or in pipeline. This would give a better idea of the success of NPs for targeting brain tumors in pediatric population.
Author’s reply: We have modified the text in the “Relevant cargoes for brain CSCs” section, according to this review’s suggestion (lines 526-541).
Reviewer 2 Report
I read the manuscript with interest.
Introduction section must be expanded in term of using nanoparticles in CSCs treatments. Authors must use this ref in the introduction section: Nanomaterials 11.7 (2021): 1755.
Authors must add table (if possible) for each section. Finally the paper must be polish by a native english speaker.
Author Response
Reviewer #2
I read the manuscript with interest.
Introduction section must be expanded in term of using nanoparticles in CSCs treatments. Authors must use this ref in the introduction section: Nanomaterials 11.7 (2021): 1755.
Author’s reply: We expanded this aspect in the introduction section (lines 135-142).
Authors must add table (if possible) for each section. Finally, the paper must be polish by a native english speaker.
Author’s reply: We thank the reviewer for the suggestion, we added a new Table 1 to summarize receptors used in NP-targeting strategy.
Reviewer 3 Report
In this review, the authors aimed to summarize novel nanotherapeutic approaches for targeting pediatric brain tumors, with a specific focus on targeting CSCs as well as other biomarkers frequently expressed in such tumors.
Overall, the review is well written and well structured and it provides interesting information in some such “hot-topics” as are nanomedicine and pediatric brain cancers, that have generally a very poor prognosis. Therefore, a state of the art of the current studies and methodologies aimed to fight these types of cancer may be useful to both basic and clinical scientists. However, there are some weaknesses that should be improved, such as a poor description of CSCs biology and the mechanisms by which nanoparticles exert their function to counteract tumors. Hence, the following commentaries should be considered before publications.
-Since the aspect of CSCs is so relevant along with the manuscript, I recommend changing the current title to have a specific mention to “nanoparticles for CSCs targeting in pediatric brain tumors “ or similar.
-In the introduction, please specify the molecular mechanisms that control CSCs in cancer and specifically in brain tumors and have a mention regarding the concept of “cell plasticity” and how it can be applied for these types of cancers, considering the following literature. PMID: 26064420, 30510501, 33640445, 35710946. Indeed, this concept is very important to fully understand cancer stem cells biology and specifically the authors should consider the role of - metabolic rewiring, - tumor microenvironment and - EMT in the regulation of brain cancer stemness. A figure may be useful to summarize these three players involved in cell plasticity. This deepening in brain cancer biology may set up the basis for a better understanding of the mechanisms by which nanoparticles act and may facilitate the reader.
- Although the review aims to summarize nanoparticle based therapies, a brief introduction regarding the current chemotherapy drugs employed in clinics (e.g. Temozolomide; radiotherapy..) and some sentences explaining why these treatments are not effective should be included. The authors should stress the weakness of standard treatments and their limitations involving nucleic acids, drugs and polymers, and highlight in this way the need for novel delivery systems.
-The authors discussed many therapeutic systems that act through different molecules. Therefore, there are many different mechanisms by which these nanodrugs may exert their effect. This may induce confusion in the reader. A figure or table to resume the main mechanisms by which these nanotherapeutics act should be provided, and it would help the reader to understand the different possible mechanisms induced by nanoparticles. For example they should organize the mechanisms leading EMT inhibition (PMID: 31861725); autophagy modulation( doi: 10.1038/s41427-022-00422-3); apoptosis (PMID: 33808150) ect.
-The authors should include some mentions of the possibility that nanoparticles can target tumor microenvironment and angiogenesis. Indeed, both of them are very important in promoting the invasiveness of glioblastoma (doi: 10.1038/s41598-022-09549-3, 10.1038/s41467-022-34208-6)
- Have a mention on the immunomodulatory effects of nanoparticles that are getting extreme importance in cancer therapy and represent fascinating therapeutic options for many cancers. 10.1016/j.bmc.2022.116913.
- There are many references that are not recent (last 5 years); considering that nanomedicine is an emerging topic, the authors should replace the oldest ones with novel more recent references.
Author Response
Reviewer #3
In this review, the authors aimed to summarize novel nanotherapeutic approaches for targeting pediatric brain tumors, with a specific focus on targeting CSCs as well as other biomarkers frequently expressed in such tumors.
Overall, the review is well written and well structured and it provides interesting information in some such “hot-topics” as are nanomedicine and pediatric brain cancers, that have generally a very poor prognosis. Therefore, a state of the art of the current studies and methodologies aimed to fight these types of cancer may be useful to both basic and clinical scientists. However, there are some weaknesses that should be improved, such as a poor description of CSCs biology and the mechanisms by which nanoparticles exert their function to counteract tumors. Hence, the following commentaries should be considered before publications.
-Since the aspect of CSCs is so relevant along with the manuscript, I recommend changing the current title to have a specific mention to “nanoparticles for CSCs targeting in pediatric brain tumors “ or similar.
Author’s reply: We modified the title according to this review’s suggestion.
-In the introduction, please specify the molecular mechanisms that control CSCs in cancer and specifically in brain tumors and have a mention regarding the concept of “cell plasticity” and how it can be applied for these types of cancers, considering the following literature. PMID: 26064420, 30510501, 33640445, 35710946. Indeed, this concept is very important to fully understand cancer stem cells biology and specifically the authors should consider the role of - metabolic rewiring, - tumor microenvironment and - EMT in the regulation of brain cancer stemness. A figure may be useful to summarize these three players involved in cell plasticity. This deepening in brain cancer biology may set up the basis for a better understanding of the mechanisms by which nanoparticles act and may facilitate the reader.
Author’s reply: We modified the text in the introduction section, and added the new Figure 1, according to this reviewer’s suggestion (lines 86-121).
- Although the review aims to summarize nanoparticle based therapies, a brief introduction regarding the current chemotherapy drugs employed in clinics (e.g. Temozolomide; radiotherapy..) and some sentences explaining why these treatments are not effective should be included. The authors should stress the weakness of standard treatments and their limitations involving nucleic acids, drugs and polymers, and highlight in this way the need for novel delivery systems.
Author’s reply: As the reviewer suggested we added some additional information on current therapies and their limitations in the introduction section (lines 51-67).
-The authors discussed many therapeutic systems that act through different molecules. Therefore, there are many different mechanisms by which these nanodrugs may exert their effect. This may induce confusion in the reader. A figure or table to resume the main mechanisms by which these nanotherapeutics act should be provided, and it would help the reader to understand the different possible mechanisms induced by nanoparticles. For example they should organize the mechanisms leading EMT inhibition (PMID: 31861725); autophagy modulation ( doi: 10.1038/s41427-022-00422-3); apoptosis (PMID: 33808150) ect.
Author’s reply: Most of the nano strategies described in the present review do not address the effects of NPs themselves but are mainly used as carrier for drug/biological molecules delivery into brain tumor stem cell compartments. For this reason, we added an additional column in New Table 2 on the different mechanisms induced by NPs-Cargo in brain tumors.
-The authors should include some mentions of the possibility that nanoparticles can target tumor microenvironment and angiogenesis. Indeed, both of them are very important in promoting the invasiveness of glioblastoma (doi: 10.1038/s41598-022-09549-3, 10.1038/s41467-022-34208-6)
Author’s reply: As the reviewer suggested we added some mentions on TME and angiogenesis and the potential application of NPs strategy, in the “Future perspectives” section (lines 759-770).
- Have a mention on the immunomodulatory effects of nanoparticles that are getting extreme importance in cancer therapy and represent fascinating therapeutic options for many cancers. 10.1016/j.bmc.2022.116913.
Author’s reply: We have added the new section “Immunotherapy and NPs in pediatric brain tumors” regarding the immunomodulatory effects of nanoparticles (lines 631-660).
- There are many references that are not recent (last 5 years); considering that nanomedicine is an emerging topic, the authors should replace the oldest ones with novel more recent references.
Author’s reply: We modified the text replacing oldest references where it was possible with more recent ones.
Round 2
Reviewer 3 Report
The authors provided adequate discussions that addressed the concerns raised by this reviewer. They included new sections, references, and rearranged text to fit my concerns. I have no more comments as I believe this manuscript has definitely improved. I thank the authors for considering my comments.